# The Molecular Basis of Male Infertility in Obesity: A Literature Review

**DOI:** 10.3390/ijms25010179

**Published:** 2023-12-22

**Authors:** Biji Thomas George, Malay Jhancy, Rajani Dube, Subhranshu Sekhar Kar, Lovely Muthiah Annamma

**Affiliations:** 1Department of Surgery, RAK College of Medical Sciences, RAKMHSU, Ras al Khaimah P.O. Box 11172, United Arab Emirates; 2Department of Pediatrics, RAK College of Medical Sciences, RAKMHSU, Ras al Khaimah P.O. Box 11172, United Arab Emirates; jhancy@rakmhsu.ac.ae (M.J.); subhranshu.kar@rakmhsu.ac.ae (S.S.K.); 3Department of Obstetrics and Gynecology, RAK College of Medical Sciences, RAKMHSU, Ras al Khaimah P.O. Box 11172, United Arab Emirates; rajani.dube@rakmhsu.ac.ae; 4Department of Clinical Sciences, Ajman University, Ajman P.O. Box 346, United Arab Emirates; l.annamma@ajman.ac.ae

**Keywords:** insulin resistance, reproductive hormone, adipokine, leptin, adiponectin, resistin, sirtuin, epigenetic, infertility, obesity, molecular basis

## Abstract

The rising incidence of obesity has coincided with rising levels of poor reproductive outcomes. The molecular basis for the association of infertility in obese males is now being explained through various mechanisms. Insulin resistance, hyperglycemia, and changes in serum and gonadal concentrations of adipokines, like leptin, adiponectin, resistin, and ghrelin have been implicated as causes of male infertility in obese males. The effects of obesity and hypogonadism form a vicious cycle whereby dysregulation of the hypothalamic–pituitary–testicular axis—due to the effect of the release of multiple mediators, thus decreasing GnRH release from the hypothalamus—causes decreases in LH and FSH levels. This leads to lower levels of testosterone, which further increases adiposity because of increased lipogenesis. Cytokines such as TNF-α and interleukins, sirtuins, and other inflammatory mediators like reactive oxygen species are known to affect fertility in obese male adults. There is evidence that parental obesity can be transferred through subsequent generations to offspring through epigenetic marks. Thus, negative expressions like obesity and infertility have been linked to epigenetic marks being altered in previous generations. The interesting aspect is that these epigenetic expressions can be reverted by removing the triggering factors. These positive modifications are also transmitted to subsequent generations.

## 1. Introduction

Overweight and obesity are terms used to describe an abnormal and excessive build-up of body fat that may have a detrimental effect on health [1]. Body mass index (BMI) is a measure of body weight per height that is widely used to classify adults as overweight or obese. It is calculated by dividing a person’s body weight (in kilograms) by the square of their height (in meters/m^2^). The World Health Organization (WHO) defines overweight as a BMI of 25 kg/m^2^ or above and obesity as a BMI of 30 kg/m^2^ or above [2]. However, these cut-offs vary by race and ethnicity. The previously mentioned cut-offs are valid for non-Asians (people of Hispanic, non-Hispanic white, non-Hispanic black, and similar race or ethnicity categories), whereas the cut-off to determine obesity in Asians is a BMI of 27.5 kg/m^2^ or above [3].

Given an obesogenic environment, excessive emphasis on low fat intake, and overindulgence in simple carbohydrates and sugar, obesity has become a global pandemic [4]. Nearly two billion adults (over 18 years of age), comprising 39% of the world’s adult population, have been reported to be overweight or obese according to WHO statistics from 2016. Of these, over 650 million fell under the obese category [2].

The rising incidence of obesity has coincided with rising levels of poor reproductive outcomes, including infertility, which are now commonly seen in obese adult males [5,6,7,8]. Men who are obese are 42% more likely than normal-weight men to have oligozoospermia and are 81% more likely to have azoospermia [9].

A wide variety of factors have been shown to influence fertility in obese persons. Some examples are endocrine factors, factors causing increases in inflammatory mediators or oxidative stress, factors that influence the effects of intracellular enzymes, and factors that produce genetic or epigenetic alterations affecting spermatogenesis and sperm function [10,11].

Understanding the molecular basis of poor reproductive outcomes in such individuals may help in better counseling such individuals and advising behavior-modifying strategies to minimize the impact of such factors [12]. Novel interventions may then target the specific responsible molecular factors to create a new frontier in preventive and therapeutic pharmacological approaches that offset poor reproductive outcomes or infertility in these individuals [13].

This article aims to highlight the current understanding of the molecular mechanisms that lead to male infertility and poor reproductive outcomes in obese male individuals.

## 2. Endocrine Changes

Insulin resistance, hyperglycemia, and adipokines have been implicated as causes of male infertility in obese males [14,15,16,17,18,19] (Figure 1).

### 2.1. Insulin Resistance and Hyperglycemia

#### 2.1.1. Insulin Resistance

The term “metabolic syndrome” encompasses a range of related conditions, including diabetes mellitus, hypertension, dyslipidemia, coronary artery disease, and obesity [19]. Insulin resistance is the inability of a known dose of exogenous or endogenous insulin to stimulate glucose uptake and metabolism in an individual to the same extent that it does in the general population in good health [20,21]. Increased insulin resistance has been noted in males with unexplained infertility [22].

Insulin resistance increases inflammation, lowers male sex hormone release, lowers sex hormone binding globulin levels, and aggravates obesity [16,20,23,24]. Insulin resistance has been shown to independently influence sperm motility and, when associated with obesity, decrease semen volume and sperm count [19,21,25].

Freshly ejaculated sperm are still without progressive motility or the capacity to fertilize. They first need to go through a process called capacitation, which entails a progression of biochemical and physiological alterations. A key requirement for fertilization is capacitation [26,27]. Insulin has been found to be secreted by sperm. It has been shown that there is a significant difference between insulin release in capacitated and non-capacitated sperm, pointing to a possible active role for insulin in capacitation. Insulin resistance could interfere with the distal effects of autocrine insulin secretion from the sperm and negatively affect capacitation [28].

Extra-pancreatic insulin receptors have been demonstrated to be present in sperm cells in mammals. The activation of these receptors has been shown to enhance sperm motility. It is also noted that the male germ cell line is capable of producing insulin itself [29,30,31]. Whether insulin resistance seen in obesity affects these receptors is not known.

#### 2.1.2. Hyperglycemia

Hyperglycemia has been shown to negatively affect male fertility, both in Type 1 and Type 2 diabetes mellitus [32,33,34]. Hyperglycemia has been demonstrated to induce testicular damage via diacylglycerol–protein kinase C and the polyol pathways in the Type 1 diabetes mellitus mouse model [35]. Advanced glycation end-products (AGE) are the result of non-enzymatic glycosylation pathways [36]. AGE receptors are expressed throughout testicular tissue. The excess binding of AGE to these receptors is postulated to play a role in the development of infertility. Hyperglycemia is also shown to affect fertility by stimulating the inflammatory pathways and increasing the production of interleukin-1 and tumor necrosis factor-α [24,32,33,35,36].

### 2.2. Adipokines

Up until the 1980s, adipose tissue was thought to be an inert reservoir of energy stored as triglycerides [5]. However, adipose tissue is now understood to be a significant endocrine organ that produces several peptide hormones called adipokines, such as leptin, adiponectin, resistin, and many others [5,37,38]. Other chemicals secreted by adipose tissue include molecules such as cytokines and chemokines [5].

Adipokines regulate a variety of physiologic functions, including hunger, metabolism, cardiovascular function, immunity, and reproduction [14]. Adipokines like leptin, adiponectin, resistin, ghrelin, orexin, and obestatin interfere with normal reproductive hormonal regulation in obese male adults [11]. It has been theorized that adipokines are significantly implicated in the pathophysiology of poor reproductive outcomes in these individuals, directly or indirectly [14]. However, some studies have failed to show any major effect of adipokines on peripheral sites in the male reproductive system [15]. The expression of various steroidogenic genes is directly impacted by adipokines [37].

#### 2.2.1. Leptin

Leptin is a non-glycosylated peptide that is largely secreted by adipocytes [13]. It is a key regulatory adipokine that plays a fundamental role in a variety of metabolic processes, including reproductive performance [11,39].

Childhood obesity has been reported in families with congenital leptin deficiency [37]. Leptin has powerful anorexigenic effects, decreasing appetite by inhibiting the orexigenic factors neuropeptide Y and agouti-related peptide and augmenting the effects of the anorexigenic peptides alpha-MSH/pro-opiomelanocortin and cocaine- and amphetamine-related transcript (CART). It also raises calorie expenditure. The Janus kinase/signal transducer and activator of the transcription complex is activated by leptin, which, in turn, affects sex hormone expression [38].

Numerous studies have reported a significant correlation between the levels of leptin, obesity, and infertility with respect to the hypothalamic–pituitary–testicular axis, the regulation of androgen levels, and the production of sperm. Elevated levels of leptin have been observed in obese infertile male individuals with disorders affecting the parenchyma of the testicles, such as nonobstructive azoospermia and oligozoospermia.

Furthermore, elevated serum leptin levels are associated with decreases in serum testosterone and sperm parameters and increases in follicle-stimulating hormone (FSH) and luteinizing hormone (LH) levels, and it can cause abnormal sperm morphology [11,13,15,37,40].

Leptin operates in a narrow range of optimum functions. Excess or deficits in the level of leptins adversely affect reproductive functions. Leptin acts directly on the kisspeptin neurons in the hypothalamus to stimulate the release of gonadotropin-releasing hormone (GnRH). Poor reproductive status, which is associated with increasing obesity, is postulated to be due to the effect of leptin on the kisspeptin-GnRH pathway [39,41].

Excessive levels of leptins are paradoxically found in obese adults [24]. A leptin resistance pathway has been hypothesized, similar to the insulin resistance pathway, whereby the excessive level of leptin has an inhibitory effect on the kisspeptin neurons, the function of the testes, and spermatogenesis [11,24,38,41].

Elevated levels of leptin also seem to weaken the nutritional function of Sertoli cells in spermatogenesis. Hyperleptinemia can impair testosterone production in the Leydig cells. It is also associated with mitochondrial dysfunction, increasing oxidative stress and affecting spermatozoal function [5].

It is probable that measuring leptin and FSH levels is a sensitive biomarker in predicting the successful retrieval of sperm in obese adult males with non-obstructive azoospermia [38].

#### 2.2.2. Adiponectin

The most abundant adipose-tissue-secreted hormone is adiponectin, which plays important functions in glucose regulation and lipid metabolism. Adiponectin has two distinctive receptors, namely, AdipoR1, which is universally expressed in skeletal muscle, and AdipoR2, which is principally expressed in the liver and adipose tissue. Both receptors are also expressed by Leydig cells, spermatozoa, and the epididymis. Adiponectin shields Leydig cells from the injurious effects of cytokines by inhibiting nuclear factor-KB signaling through the stimulation of testicular 5′ AMP-activated protein kinase [38].

The production of adiponectin by adipose tissue increases during starvation and is inversely related in obese individuals to visceral adiposity and BMI [1,37,38]. Furthermore, adiponectin increases insulin sensitivity. Decreases in adiponectin levels result in insulin resistance and metabolic syndrome. Adiponectin levels have been found to be low in obese females [1].

Adiponectin has been found in circulation in various molecular isoforms. Four-fifths of circulating adiponectin is the high-molecular-weight isoform. The level of adiponectin is nearly double in serum when compared with that in human seminal plasma. This is hypothesized to be due to the blood–testis barrier. The measured adiponectin level in semen is lower in overweight and obese patients when compared with patients with normal BMIs [5,42,43].

The level of circulating adiponectin is inversely proportional to the level of circulating testosterone [44]. Possibly because of the low serum concentrations of testosterone seen in obese males, levels of serum adiponectin have been shown to be paradoxically elevated in obese males. This suggests a resistance phenomenon similar to insulin and leptin in obese males [11].

#### 2.2.3. Resistin

Resistin is a polypeptide that is involved in the development of insulin resistance. The modulation of insulin sensitivity and the differentiation of adipocytes are among the known functions of resistin [37]. Increased concentrations of resistin have been correlated with low sperm motility and vitality. However, other studies have not shown any effect of resistin on sperm function [43].

Resistin is also associated with increased levels of cytokines such as interleukin-6 and elastase in seminal fluid and is implicated in the inflammation of the reproductive organs noted in obese males [5,45]. Apart from the peripheral effects of resistin on the reproductive system in obese males, resistin has also been isolated from the pituitary gland and hypothalamus in humans, suggesting a central action too [5,46].

#### 2.2.4. Ghrelin

Ghrelin is an orexigenic peptide released from the stomach during fasting states and has been called the “hunger hormone” [11,13]. Circulating levels of ghrelin have been shown to be inversely correlated with BMI. When administered centrally, ghrelin has been shown to increase the utilization of glucose in adipose tissue. It has been shown to protect the testes and sperm from endoplasmic reticulum stress and inhibit the inflammatory response to stress and apoptosis [47].

Ghrelin in association with leptin is involved in the control of adiposity in obese individuals. Ghrelin has been found in Leydig and Sertoli cells in the testes and inhibits testosterone secretion. It also plays a role in the regulation of spermatogenesis, and decreased levels may suppress the secretion of FSH and LH in adult men [13,48,49,50,51]. Low serum concentrations of ghrelin have been found to adversely decrease semen volume, sperm motility, and morphology [52].

#### 2.2.5. Chemerin

Chemerin is another peptide secreted by adipocytes that modulates insulin [5]. Concentrations of chemerin are elevated in obese individuals when compared with normal-weight individuals, and its levels are directly correlated with increases in BMI and inversely correlated with levels of testosterone [43]. Chemerin and its receptors have been found in the testes and adipose tissue and have adverse effects on testicular steroidogenesis. Chemerin effects the release of gonadotropins from the hypothalamus and is concentrated in the Leydig cells. Anti-androgens have been demonstrated to inhibit the expression of chemerin in Leydig cells [53,54,55]. Adiponectin and chemerin are called “contrary adipokines”, as they display opposing functions, where adiponectin promotes insulin sensitivity while chemerin inhibits the signaling of insulin. A high-adiponectin-to-low-chemerin ratio is suggested to play an important role in the development of metabolic syndrome in obesity, and this might be a target of therapeutic interventions meant to treat infertility in obese males [56].

#### 2.2.6. Visfatin

Visfatin, which is also known as a pre-B-cell colony-enhancing factor or phosphoribosyltransferase, is a protein whose role in obesity is uncertain [5]. It has been theorized to be an insulin-like adipocytokine secreted from adipose cells. Protective and adverse roles in obese individuals have both been reported. More recently, it has been reported to produce proinflammatory expression in genes, including the *CD68* and *tumor necrosis factor-alpha (TNF-α) genes* in adipose tissue, and is associated with insulin resistance and hyperlipidemia [57,58,59]. Visfatin is found in seminal fluid, Leydig cells, and spermatozoa and has been shown to increase testosterone levels [60,61].

#### 2.2.7. Apelin, Omentin, Hepcidin, and Vaspin

Apelin increases with adiposity and with increased insulin levels. It modulates insulin resistance by increasing the expression of uncoupling proteins in adipose tissue and decreasing the levels of adiponectin [62,63]. Omentin-1 seems to have an anti-inflammatory and antioxidative action [64]. Levels of omentin-1/intelectin, like adiponectin, decrease with insulin resistance, BMI, and adiposity [62,65]. Hepcidin seems to modulate inflammatory cytokines like TNF-α and C-reactive protein, and its concentrations have been found to increase in obese individuals [66]. Vaspin is a serine protease inhibitor that is also an insulin sensitizer and has increased concentrations in adipose individuals, especially those with increased glucose tolerance [62].

### 2.3. Reproductive Steroids

#### 2.3.1. Hypothalamic–Pituitary–Testicular Axis

Normal reproductive hormone secretion is imperative for normal male fertility. Infertility may result from any interference with the intricately balanced interplay between the constituents of the hypothalamic–pituitary–testicular axis [67,68].

The hypothalamic–pituitary–testicular axis regulates the production of testosterone, which, in turn, is influenced by direct negative feedback from testosterone. LH and FSH are secreted in response to the pulsatile release of gonadotropin-releasing hormone (GnRH). LH stimulates testosterone secretion by acting on Leydig cells, and FSH induces spermatogenesis by acting on Sertoli cells in the testes [5,7].

The effects of obesity and hypogonadism form a vicious cycle, whereby dysregulation of the hypothalamic–pituitary–testicular axis—due to the effect of the release of multiple mediators, thus decreasing GnRH release from the GnRH neurons of the hypothalamus—causes decreases in LH and FSH levels. This leads to lower levels of testosterone, which further increases adiposity because of increased lipogenesis [69].

Since the 1970s, researchers have identified hormonal differences between obese and non-obese adult males in the hypothalamic–pituitary–testicular axis. Adult obese males have been found to have decreased peripheral levels of total testosterone and sex hormone-binding globulin (SHBG). Lower levels of SHBG are mostly mediated by elevated levels of circulating insulin linked to the insulin resistance of obesity [25]. The significantly decreased SHBG concentration is the primary cause of the concurrent decrease in total testosterone [70]. There is a negative correlation between the degree of obesity, as measured by the BMI, and levels of total testosterone and SHBG [7,17,67,71,72]. The typical reproductive hormonal profile in obese adult males is that of hyperestrogenic hypogonadotropic hypoandrogenemia [25].

#### 2.3.2. Estrogens

The peripheral aromatization of androgens leads to elevated levels of estrogens. Increased adiposity in obese males boosts conversion rates of testosterone into estradiol [11,25,37]. Estradiol blunts GnRH pulses in the hypothalamus and, therefore, FSH and LH secretion in the pituitary. The rise in estradiol levels in obese men reduces the production of FSH and LH, which impairs testicular function and lowers levels of testosterone [25].

Kisspeptin neurons in the hypothalamus act as a center for the regulation of reproductive hormones. Numerous regulatory factors govern the release of kisspeptin from these neurons, which, in turn, affects the release of GnRH from GnRH neurons that control the reproductive axis. Estrogens produce negative feedback in the arcuate nucleus of the hypothalamus, which specifically decreases the production of kisspeptin, thereby inhibiting the pulsatile release of GnRH, resulting in decreased production of FSH and LH from the pituitary gland [11,73,74].

#### 2.3.3. Inhibin B

Inhibin B is a protein dimer composed of an alpha and a beta subunit produced primarily by Sertoli cells in the prepubertal testes. The site of secretion in adult males is still not clear. Inhibin B regulates FSH production via a negative feedback mechanism and, in turn, is regulated by FSH and LH levels. There is a positive correlation between testicular volume and sperm production with the level of circulating inhibin B. This makes the measurement of inhibin B a good marker of male reproductive function [75,76,77]. Inhibin B is considered to be a marker for “nurse” Sertoli cells, which are primarily involved in the nutrition and physical support of spermatogenesis [13].

### 2.4. Incretins

Glucagon-like peptide-1 (GLP-1), which is an incretin hormone, is responsible for the control of insulin release from the pancreatic islet in response to meal ingestion and consequent glucose loading. It has a very short half-life of 1.5 to 5 min and is rapidly cleared by the kidney [78]. It crosses the blood–brain barrier and induces satiety, thereby reducing food intake. Low levels of GLP-1 have been shown to reduce the testosterone pulse frequency and reduce testicular weight [13]. The administration of GLP-1 receptor antagonists has been demonstrated to increase testosterone levels and increase testicular weight [79].

## 3. Inflammatory Molecules

Obesity is associated with low-grade, chronic systemic inflammation. The pathophysiology seems to comprise changes in immune parameters, especially in TH1-lymphocytes and M1 macrophages. The chronic inflammation associated with obesity causes insulin resistance and metabolic syndrome [62,80]. Lower testosterone levels have been demonstrated in obese individuals who have high levels of inflammatory markers [81,82]. Other effects of inflammatory molecules are shown in Figure 2.

### 3.1. C-Reactive Protein

Concentrations of the inflammatory marker known as high-sensitivity C-reactive protein have been found to be positively correlated with an increasing BMI. This is also associated with higher concentrations of serum insulin in obese males. The chronic inflammation associated with obesity causes insulin resistance and metabolic syndrome [80]. This suggests that metabolic syndrome may cause inflammatory changes, which then lead to subfertility in obese male adults [24]. The increased production of inflammatory mediators in conditions like burns and sepsis is associated with low testosterone levels [83,84]. Similarly, epidemiological studies have demonstrated lower testosterone levels in obese individuals who have high levels of inflammatory markers like C-reactive protein [81,82].

### 3.2. Reactive Oxygen Species

Several factors are ubiquitous in obesity, like hyperglycemia, hyperleptinemia, and mitochondrial dysfunction, and are known to produce increased levels of reactive oxygen species (ROS) [69,85,86]. ROS are implicated as an influential factor for up to four-fifths of all cases of male factor infertility [87]. They cause oxidative stress in spermatozoa. The most important of these species are nitric oxide, hydroxyl radicals, and superoxide anions. Major contributors to inflammation-related damage to spermatozoa are reactive oxygen species [88]. The two most important sources of ROS in semen are leukocytes and sperm. The damage to spermatozoa is caused by two primary mechanisms: damage to the sperm membrane and damage to sperm mitochondrial DNA [87]. Oxidative stress has been implicated in the induction of male infertility in obese mice [89].

### 3.3. Cytokines

Adipocytes produce a large number of cytokines such as TNF-α and interleukins (ILs). These cytokines, especially IL-6, are implicated in the insulin resistance that is often seen in obese males. These pro-inflammatory cytokines are seen in increased levels in the semen, testicular parenchyma, and circulating blood of obese males. Alterations in the junctional proteins caused by TNF-α and IL-6 between adjacent Sertoli cells may cause impaired spermatogenesis because of changes in the structure of the seminiferous epithelium. Cytokines also attract macrophages and neutrophils to the epididymal lumen, which affects sperm maturation and function [57,88].

In addition to peripheral action at the gonadal level, these cytokines also have a central action at the hypothalamic–pituitary–testicular axis by inhibiting LH, which, in turn, leads to lower testosterone levels and subfertility in obese male adults [88].

The inability to maintain the adequate erection of the penis to achieve sexual satisfaction is termed erectile dysfunction. An increase in cavernosal pressure due to vasodilation caused by nitric oxide (NO) from the action of nitric oxide synthase (NOS) leads to the phenomenon of erection. TNF-α acts on endothelial cells, causing the generation of ROS intracellularly, which prevents the generation of NO and, therefore, vasodilation. This action along with its inhibitory effect on NOS causes erectile dysfunction [80].

### 3.4. Sirtuins

Sirtuins (SIRTs) are conserved nicotinamide adenine dinucleotide (NAD+)-dependent deacetylases associated with cellular metabolism. There are seven types of SIRTs, from SIRT1 to SIRT7, of which SIRT1 is linked to obesity and controls the genetic expression of insulin secretion and sensitivity and adipose tissue formation and facilitates inflammatory responses [90].

SIRT1 levels are decreased in adipose tissue in obese individuals, but when inappropriately stimulated, they are implicated in the inflammatory response in chronically hypoxic peripheral adipose tissue in the obese [91,92]. SIRT1 has also been identified in the testes and all phases of sperm maturation, which suggests a role in modulating spermatogenesis through its regulatory effect on glycosylation and the formation of lactate in Sertoli cells, which are the primary substrates of germ cells. Lower levels of SIRT1 have been shown to lead to arrested spermatogenesis because of mitochondrial dysfunction caused by the dysregulation of peroxisome proliferator-activated receptor-gamma coactivator (PCG-1α), which is a key element in mitochondrial processes [5,93,94].

SIRT1 has a role in protecting reproductive cells from oxidative stress triggered by ROS. Decreased levels of SIRT1 and SIRT3 are inversely correlated with levels of ROS [95].

SIRT1 has also been shown to be present in increased levels in the human hypothalamus, particularly in the GnRH neurons. Decreased levels in obese male individuals may disrupt the hypothalamus–pituitary–testicular axis, causing lower FSH and LH levels and, therefore, lower levels of testosterone [5].

SIRTs are also involved in lipid metabolism, especially SIRT3, which is involved in the oxidation of fatty acids. Reduced levels of SIRT3 may affect steroidogenesis. SIRT3 is also involved in the regulation of adenosine triphosphate production in the mitochondria of spermatozoa and low levels may affect spermatogenesis [93].

## 4. Intracellular Factors

### 4.1. Fatty Acids and Male Subfertility

Fatty acids can be saturated, such as palmitic and myristic acids, or unsaturated, such as oleic acid and linoleic acid. Both saturated and unsaturated fatty acids are found in the membrane of spermatozoa. This composition is critical for proper sperm function. Increases in the BMI associated with inflammatory changes and oxidative stress alter the lipid composition of the sperm membrane. These changes in the composition of fatty acids in spermatozoa may be the root cause of subfertility in obese male adults [88,96].

### 4.2. Cholesterol and Sperm Capacitation

In addition, the efflux of cholesterol in sperm membranes during capacitation contributes to sperm motility and function [88]. This occurs early in the process of capacitation and involves a decrease in the cholesterol–phospholipid ratio in sperm membranes. A subsequent cascade of reactions involving adenyl cyclase, cyclic adenosine monophosphate, and protein kinase-A results in the phosphorylation of sperm protein tyrosine residue and enables the hyperactive motility of sperm and their capacity to bind to the zona pellucida in order to proceed with the acrosomal reaction. High levels of cholesterol have been found in the spermatozoa of obese males, which may also contribute to decreased sperm function and premature acrosome reaction. Cholesterol has also been shown to have adverse effects on the epididymis, ejaculate volume, and morphological abnormalities in sperm [88,97].

## 5. Obesity and Epigenetic Alterations

The traditional investigation of male infertility has involved semen analysis. However, semen analysis is a poor predictor of fertility in males. In fact, adverse semen analysis results have been found in infertile and fertile males [80,98].

The relative telomere length in spermatozoa has been shown to be shorter in obese male individuals when compared with normal-weight individuals. The shortening is inversely proportional to the BMI. Percentages of the DNA fragmentation index, the maturation index, and immature chromatin are found at increased levels in the sperm of obese adult males [88,99].

Sperm DNA integrity is protected by the replacement of histones in the spermatozoa with protamine via histone acetylation. Histone acetylation has been shown to be impaired in male obesity, which results in increased damage to sperm DNA [88].

The consequences of these changes are a significant decrease in semen volume, sperm count, the progressive motility of sperm, and viability in adult males who are overweight or obese [25,100].

There is evidence that parental obesity is transferred through subsequent generations to offspring [101,102]. Such a transfer of phenotypical traits is not necessarily due to genetic transfer via mutated DNA but could also be due to epigenetic alterations. Epigenetics is the alteration of the expression of genes caused by changes in the chromatin structure of genes without modifications of their nucleotide sequences [5,101]. Environmental factors, such as obesity, diet, exposure to environmental toxins, and aging, to list a few, can switch epigenetic marks on and off, thereby influencing the expression of genes without DNA mutations. These signatures work through three different pathways: DNA methylation, histone modifications of nuclear proteins, and how non-coding mRNA is expressed [12]. These gene expressions are transmitted across subsequent generations. Thus, negative expressions like obesity and infertility have been linked to epigenetic marks being altered in previous generations. The interesting aspect is that these epigenetic marks can be switched off too by removing the triggering factors. These positive modifications also transmit to subsequent generations [102,103,104,105].

## 6. Conclusions

The rising incidence of obesity has coincided with rising levels of poor reproductive outcomes [5,6,7,8]. A wide variety of factors has been shown to influence fertility in obese persons [11,80].

Insulin resistance, hyperglycemia, and adipokines have been implicated as causes of male infertility in obese males [14,15,16,17,18,19]. Numerous studies have reported a significant correlation between the levels of leptin, obesity, and infertility with respect to the hypothalamic–pituitary–testicular axis, decreases in serum testosterone and sperm parameters, and increases in follicle-stimulating hormone (FSH) and luteinizing hormone (LH) levels, as well as abnormal sperm morphology [11,13,15,37,40]. Levels of serum adiponectin have been shown to be paradoxically elevated in obese males. This suggests a resistance phenomenon similar to insulin and leptin in obese males [11]. Resistin is associated with increased levels of cytokines in seminal fluid and has been implicated in the inflammation of the reproductive organs noted in obese males [5,45]. Low serum concentrations of ghrelin have been found to adversely decrease semen volume, sperm motility, and morphology [52]. Chemerin and its receptors have been found in the testes and adipose tissue and have adverse effects on testicular steroidogenesis [53,54]. Visfatin is found in seminal fluid, Leydig cells, and spermatozoa and has been shown to increase testosterone levels [60,61].

The effects of obesity and hypogonadism form a vicious cycle whereby dysregulation of the hypothalamic–pituitary–testicular axis—due to the effect of the release of multiple mediators, thus decreasing GnRH release from the GnRH neurons of the hypothalamus—causes decreases in LH and FSH levels. This leads to lower levels of testosterone, which further increases adiposity because of increased lipogenesis [69]. The peripheral aromatization of androgens leads to elevated levels of estrogens. Increased adiposity in obese males boosts the conversion rates of testosterone to estradiol [11,25,37]. Estrogens produce negative feedback on the arcuate nucleus of the hypothalamus, which specifically decreases the production of kisspeptin, thereby inhibiting the pulsatile release of GnRH, resulting in the decreased production of FSH and LH from the pituitary gland [11,73,74]. There is a positive correlation between testicular volume and sperm production with the level of circulating inhibin B. This makes the measurement of inhibin B a good marker of male reproductive function [75,76,77]. Low levels of GLP-1 have been shown to reduce the testosterone pulse frequency and reduce testicular weight [13].

Obesity is associated with low-grade, chronic systemic inflammation. [62,80]. The chronic inflammation associated with obesity causes insulin resistance and metabolic syndrome [80]. This suggests that metabolic syndrome may cause inflammatory changes, which then lead to subfertility in obese male adults [24]. Several factors are ubiquitous in obesity, like hyperglycemia, hyperleptinemia, and mitochondrial dysfunction, and are known to produce increased levels of reactive oxygen species (ROS) [69,85,86]. ROS have been implicated as an influential factor in up to four-fifths of all cases of male factor infertility [87]. Adipocytes produce a large number of cytokines such as TNF-α and interleukins (ILs). These pro-inflammatory cytokines are seen in increased levels in the semen, testicular parenchyma, and circulating blood of obese males and cause impaired spermatogenesis because of changes in the structure of the seminiferous epithelium. Cytokines also attract macrophages and neutrophils to the epididymal lumen, which affects sperm maturation and function [57,88]. In addition to peripheral action at the gonadal level, these cytokines also have a central action at the hypothalamic–pituitary–testicular axis by inhibiting LH, which, in turn, leads to lower testosterone levels and subfertility in obese male adults [88]. TNF-α acts on endothelial cells, causing the generation of ROS intracellularly, which prevents the generation of NO and, therefore, vasodilation. This action, along with its inhibitory effect on NOS, causes erectile dysfunction [80]. Sirtuin-1 levels are decreased in adipose tissue in obese individuals, which has been shown to lead to arrested spermatogenesis because of mitochondrial dysfunction caused by the dysregulation of PCG-1α, which is a key element in mitochondrial processes. [5,93,94] and is inversely correlated with levels of ROS [95].

Both saturated and unsaturated fatty acids are found in the membrane of spermatozoa. Increases in the BMI associated with inflammatory changes and oxidative stress alter the lipid composition of the sperm membrane. These changes in the composition of fatty acids in spermatozoa may be the root cause of subfertility in obese male adults [88,96]. High levels of cholesterol have been found in the spermatozoa of obese males, which also may contribute to decreased sperm function and premature acrosome reaction. Cholesterol has also been shown to have adverse effects on the epididymis, ejaculate volume, and morphological abnormalities in sperm [88,97].

Sperm DNA integrity is protected by the replacement of histones in the spermatozoa with protamine via histone acetylation. Histone acetylation has been shown to be impaired in male obesity, which results in increased damage to sperm DNA [88], causing a significant decrease in semen volume, sperm count, the progressive motility of sperm, and viability in adult males who are overweight or obese [25,100]. There is evidence that parental obesity is transferred through subsequent generations to offspring [101,102]. Epigenetic marks influence the expression of genes without DNA mutations. These gene expressions are transmitted across subsequent generations. Thus, negative expressions like obesity and infertility have been linked to epigenetic marks being altered in previous generations. The interesting aspect is that these epigenetic marks can be switched off too by removing the triggering factors. These positive modifications also transmit to subsequent generations [102,103,104,105].

Understanding the molecular basis of poor reproductive outcomes in such individuals may help in better counseling such individuals and advising behavior-modifying strategies to minimize the impact of such factors. Just as it is now evident that poor reproductive outcomes in women are linked to obesity, it may become clear that obese men benefit from weight loss before attempting conception [12,106]. There is a very limited number of trials using humans proving that the multiple factors in obese males cited in this review do, in fact, cause infertility; more robust trials are required to verify causation. Well-designed and adequately powered randomized trials are also required to quantify the improvement in reproductive outcomes caused by weight reduction strategies in obese males and to ascertain if such enhanced reproductive outcomes also result in improvements in such outcomes regarding their progeny.

## Figures and Tables

**Figure 1 ijms-25-00179-f001:**
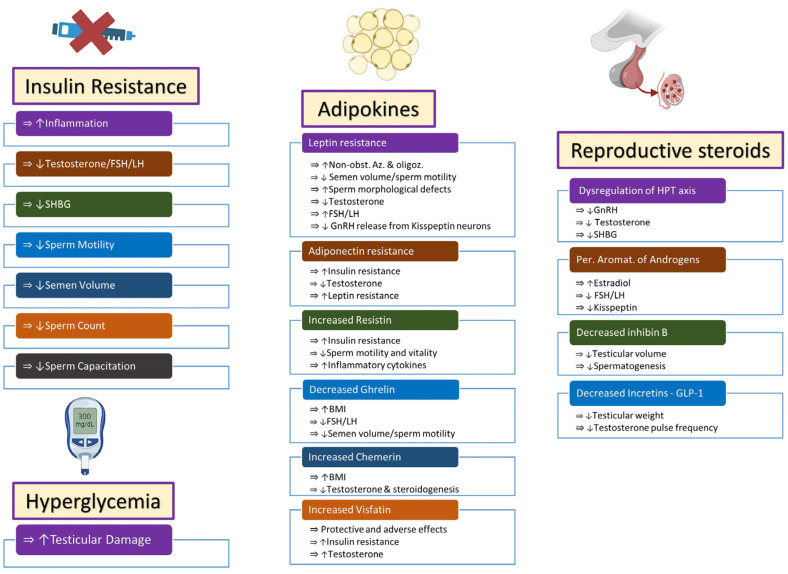
Endocrine changes in obese males lead to infertility via multiple pathways. Increased insulin resistance and hyperglycemia affect fertility via effects on inflammatory molecules, steroidogenesis, and semen quality. Adipokines behave like endocrine organs and secrete multiple hormones that have protective or adverse effects on semen and sperm quality, steroidogenesis, insulin resistance, and inflammatory mechanisms. Dysregulation of steroidogenesis occurs and leads to alterations in steroidogenesis through central and peripheral pathways. Symbols: (⇒): is associated with/is correlated with; (↑): increased; (↓): decreased. Abbreviations: FSH: follicle-stimulating hormone; LH: luteinizing hormone; SHBG: sex hormone-binding globulin; Non-obst. Az: non-obstructive azoospermia; Oligoz: oligozoospermia; GnRH: gonadotropin-releasing hormone; BMI: body mass index; HPT axis: hypothalamus–pituitary–testicular axis; Per. Aromat.: peripheral aromatization; GLP-1: glucagon-like peptide-1.

**Figure 2 ijms-25-00179-f002:**
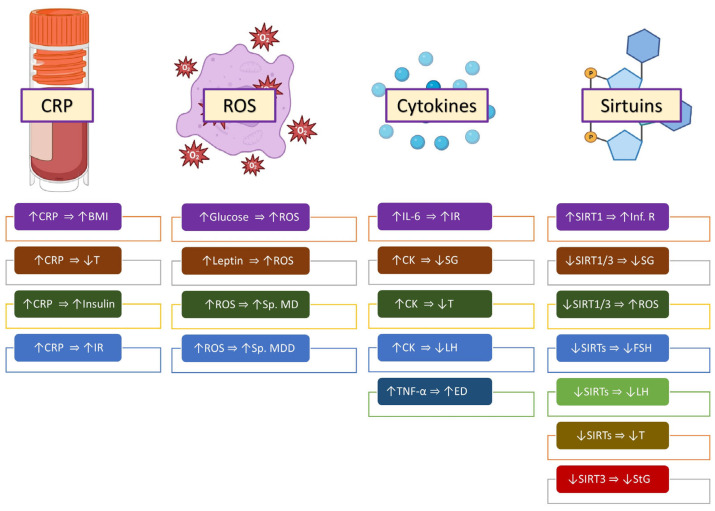
Numerous inflammatory markers and molecules are associated with male obesity. The molecular mechanisms by which these markers and molecules are correlated with obesity, steroidogenesis, and insulin resistance, as well as semen and sperm quality, are shown graphically. Symbols: (⇒): is associated with/is correlated with; (↑): increased; (↓): decreased. Abbreviations: CRP: C-reactive protein; BMI: body mass index; T: testosterone; IR: insulin resistance; ROS: reactive oxygen species; Sp. MD: sperm membrane damage; Sp. MDD: sperm mitochondrial DNA damage; IL-6: interleukin-6; CK: cytokines; SG: spermatogenesis; LH: luteinizing hormone; TNF-α: tumor necrosis factor-α; ED: erectile dysfunction; SIRT: sirtuin; FSH: follicle-stimulating hormone; StG: steroidogenesis.

## Data Availability

No new data were created or analyzed in this study. Data sharing does not apply to this article.

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
