# Peer review of "The Molecular Basis of Male Infertility in Obesity: A Literature Review"

_ijms, 2023, doi:10.3390/ijms25010179_

Round 1
Reviewer 1 Report (Previous Reviewer 2)
Comments and Suggestions for Authors
The George et al., 2023, Manuscript ID: ijms- 2707065 addresses the information on obesity-associated male infertility and its molecular mechanisms. A search on Pubmed.gov for the terms "Male Infertility" and "Obesity" keywords resulted in so many hits, nearly 183 review articles published on this topic that depict how important this topic is and how significant the problem is associated with human fertility-associated outcomes. There are a few important queries and a few suggestions that make this manuscript more representative to be published.
1. The role of obesity-associated hormones plays a big role in male infertility, but the authors fail to cite the number of previously published research articles on a similar topic that is closely associated with the mechanism of obesity-associated male infertility. The authors must cite these obesity hormones and male infertility articles PMID: 31676315
2. The article could be more explicit in distinguishing between causation and correlation. While various factors associated with male infertility in obesity are mentioned, the strength of evidence supporting causation should be discussed.
3. Some statements lack specific citations to the original studies or reviews, making it challenging to verify the information presented. Complete and accurate citations should be provided for each claim made.
4. The authors should include aging-associated male infertility too for one of the possible mechanisms.
5. The knowledge cutoff date is not provided, and the information may become outdated as new studies are published.
6. There is no mention of a systematic search for studies, and there is no discussion about the potential for publication bias. A comprehensive literature search strategy and an evaluation of publication bias are critical components of a robust systematic review.
7. The authors should consider adding another important adipokine which aims to demonstrate the recuperative effect on testicular dysfunction in diet-induced obesity (must cite and include in the literature in molecular mechanisms, PMID: 31691259.
Author Response
The George et al., 2023, Manuscript ID: ijms- 2707065 addresses the information on obesity associated male infertility and its molecular mechanisms. A search on Pubmed.gov for the terms "Male Infertility" and "Obesity"keywords resulted in so many hits, nearly 183 review articles published in this topic that depicts the how important this topic is and how significant the problem is associated with human fertility associated outcomes. There are few important queries and few suggestions which makes this manuscript more representable to be publish.
The authors are obliged to the reviewer for the reiteration of the importance of this topic. We are also indebted for the constructive comments which we have tried our best to address.
- The role of obesity-associated hormones plays a big role in male infertility, but the authors fail to cite the number of previously published research articles on a similar topic that is closely associated with the mechanism of obesity-associated male infertility. The authors must cite these obesity hormones and male infertility articles PMID: 31676315
The authors recognize the reviewer for this constructive comment. However, it may be noted that numerous studies mentioning the role of obesity associated hormones and the effect on male infertility have been included in the reference list. Reference numbers 5, 6, 11, 13, 14, 15, 16, 37, 38, 39-83 deal with this topic and are referred to at various places in the manuscript.
The suggested article PMID: 31676315 refers to the protective role of adiponectin against testicular impairment in high-fat diet/streptozotocin-induced type 2 diabetic mice which is not the subject matter of this manuscript. The protective effect of adiponectin is not demonstrated in obese humans, possibly due to a resistance effect as mentioned in lines 189-190 of the manuscript.
Hence the authors respectfully submit that this reference should not be included in the manuscript.
- The article could be more explicit in distinguishing between causation and correlation. While various factors associated with male infertility in obesity are mentioned, the strength of evidence supporting causation should be discussed.
The authors are gratified for this comment. Please note that we have answered this point during the first round of review. We have tried to distinguish between causation and correlation to the extent current knowledge allows. Since there are limited studies in humans (which are the subject of our review), more robust trials are required to verify causation. The authors have added this disclaimer to the conclusion section.
- Some statements lack specific citations to the original studies or reviews, making it challenging to verify the information presented. Complete and accurate citations should be provided for each claim made.
The authors thank the reviewer for this critical comment. Please note that we have answered this point during the first round of review. Some of the mentioned studies have been added to the manuscript and highlighted in fluorescent green. However, we humbly submit that enough original articles and reviews on this subject have already been cited to support our claims.
- The authors should include aging-associated male infertility too for the one of the possible mechanism.
The authors acknowledge the reviewer for this well-taken comment. Please note that we have answered this point during the first round of review. However aging-associated male infertility is out of the scope of this article and hence is not included as a mechanism in obesity related infertility.
- The knowledge cutoff date is not provided, and the information may become outdated as new studies are published.
The authors recognize the validity of this comment. Please note that we have answered this point during the first round of review. However this holds true for all published articles. The article is expected to be published in the near future, hence the authors do not anticipate that the information will be outdated in this interval.
- There is no mention of a systematic search for studies, and there is no discussion about the potential for publication bias. A comprehensive literature search strategy and an evaluation of publication bias are critical components of a robust systematic review.
The authors thank the reviewer for this comment. Please note that we have answered this point during the first round of review. We wish to humbly note that this article is not meant to be a systematic review, rather it is a descriptive review of the current knowledge as regards obesity and infertility in humans. As such we have minimized the reference to studies in other species as those findings may not translate well to humans. In future, the authors plan to publish a systematic review on one of the many specific sub-facets highlighted in this review article.
- The authors should consider adding another important adipokine which aims to demonstrate the recuperative effect on testicular dysfunction in diet-induced obesity (must cite and include in the literature in molecular mechanisms, PMID: 31691259."
The authors thank the reviewer for this comment and suggested inclusion of this adipokine. Nesfatin-1 has been shown to ameliorate testicular injury due to various causes but there are no well-constructed clinical studies which demonstrate its role in infertility caused by male obesity. This may be a good subject for a future manuscript.
The suggested article PMID: 31691259 is titled "Nesfatin-1 ameliorates type-2 diabetes-associated reproductive dysfunction in male mice" and does not fit in with subject of this manuscript which is infertility associated with male obesity. As such the authors respectfully submit that this reference is not suitable for this manuscript.
(Details of the first round of review are attached.)

Reviewer 2 Report (Previous Reviewer 1)
Comments and Suggestions for Authors
Thanks to the authors for having addressed all the queries appropriately. The manuscript looks improved.
Comments on the Quality of English LanguageI still suggest that a thorough editing and language-check should be performed.
For instance
Line 206: and to inhibit, not inhibits
line 220 chemerin affects, not effects
line 511 a preposition between the two sentences is lacking
Author Response
The authors are grateful to the reviewer for the kind words and encouragement. We are also grateful for the constructive comments for the previous review from the respected reviewer which we have tried our best to address.
(Details of the first round of review are attached.)

This manuscript is a resubmission of an earlier submission. The following is a list of the peer review reports and author responses from that submission.
Round 1
Reviewer 1 Report
Comments and Suggestions for Authors
The present review aims to recapitulate the molecular bases of infertility in obesity. Hence, it takes into account several aspects which are dysregulated when obesity or overweight ensue. The work is comprehensive enough, however, while some aspects are very well discussed, a few of them should deserve more space. In particular:
- the role of insulin production and expression by sperm cells, in the light of the more recent evidence (the only study cited by the authors in this points dates back to 2005).
- the role of hyperglycemia in male infertility. Please devote a couple of lines more to this aspect.
- line 183 Resistin is also associated with increased levels of cytokines in seminal fluid: which cytokines?
- lines 196-198: which is the putative mechanism through which ghrelin interferes with sperm?
- line 204: how can chemerin affect steroidogenesis?
- line 275: It has a very short half-life: how long?
The conclusion is redundant, being just a summary of what has been previously said. I suggest that the future directions of research and practical challenges of understanding and treating infertility in obese patients should be the focus of this last paragraph (expand lines 481-483).
Finally, there are some minor issues. A few examples:
Line 35: overweight and obesity are not the same (preposition “or” is incorrect)
Lines 39-40 mind that this is not applicable to every ethnicity.
Comments on the Quality of English LanguageA thorough editing of the manuscript should be done. A few examples:
Line 23 and 238: increase; line 68: affect
Line 242: the acronym SHBG should have been introduced previously in the text (paragraph 2.1), while BMI (line 247) has already been introduced
Line 278: has, not have
Line 319: are, not is
Line 364: alter
Reviewer 2 Report
Comments and Suggestions for Authors
The George et al., 2023, Manuscript ID: ijms- 2707065 addresses the information on obesity associated male infertility and its molecular mechanisms. A search on Pubmed.gov for the terms "Male Infertility" and "Obesity"keywords resulted in so many hits, nearly 183 review articles published in this topic that depicts the how important this topic is and how significant the problem is associated with human fertility associated outcomes. There are few important queries and few suggestions which makes this manuscript more representable to be publish.
1. The article could be more explicit in distinguishing between causation and correlation. While various factors associated with male infertility in obesity are mentioned, the strength of evidence supporting causation should be discussed.
2. Some statements lack specific citations to the original studies or reviews (37709866, 37140983), making it challenging to verify the information presented. Complete and accurate citations should be provided for each claim made. The role of obesity-associated hormones plays a big role in the male infertility but the authors fail to cite the number of review and research article on the similar topic which is closely associated with the mechanism of obesity-associated male infertility. The authors must cite these obesity-hormones and male infertility articles PMID: 29669464.
3. The authors should include aging-associated male infertility too for the one of the possible mechanism.
4. The knowledge cutoff date is not provided, and the information may become outdated as new studies are published.
5. There is no mention of a systematic search for studies, and there is no discussion about the potential for publication bias. A comprehensive literature search strategy and an evaluation of publication bias are critical components of a robust systematic review.
6. More graphical representations of molecular mechanisms “adipokines” and “insulin resistance” should be made.